# Mastering Yield Stress Evolution and Formwork Friction for Smart Dynamic Casting

**DOI:** 10.3390/ma13092084

**Published:** 2020-05-01

**Authors:** Anna Szabo, Lex Reiter, Ena Lloret-Fritschi, Fabio Gramazio, Matthias Kohler, Robert J. Flatt

**Affiliations:** 1Institute of Technology in Architecture, ETH Zurich, Stefano-Franscini-Platz 1, 8093 Zürich, Switzerland; szabo@arch.ethz.ch (A.S.); lloret@arch.ethz.ch (E.L.-F.); gramazio@arch.ethz.ch (F.G.); kohler@arch.ethz.ch (M.K.); 2Institute for Building Materials, ETH Zurich, Stefano-Franscini-Platz 3, 8093 Zürich, Switzerland; reiter@ifb.baug.ethz.ch

**Keywords:** smart dynamic casting, set on demand, accelerator, process window, SDC number

## Abstract

The construction industry is a slow adopter of new technologies and materials. However, interdisciplinary research efforts in digital fabrication methods with concrete aim to make a real impact on the way we build by showing faster production, higher quality and enlarged freedom of design. In this paper, the potential and constraints of a specific digital slip-forming process, smart dynamic casting (SDC), are investigated with a material-focused approach in the complex task of producing thin folded structures. Firstly, the workability and the strength evolution of different material compositions are studied to achieve the constant processing rate for SDC. Secondly, friction between the formwork walls and the concrete, a key aspect in slip-casting, is studied with a simplified experimental setup to identify if any of these mixes would provide an advantage for processing. Finally, a theoretical framework is constructed to link the material properties, the process conditions and the designed geometry. This framework introduces the ‘SDC number’ as a simplified approach to formulate the process window, the suitable conditions for slip-forming. The experimental results prove the assumption of the model that friction is proportional to yield stress for all base compositions and acceleration methods regardless of the filling history. The results are evaluated in the context of the narrow process window of thin folded structures as well as the wider process window of columns. The necessity of consistent strength evolution is underlined for narrow windows. Further, friction is shown to be the highest initially, thus with both narrow and wide process windows, after a successful start-up the continuation of slipping is less prone to failure. The proposed theoretical model could provide material and geometry-specific slipping strategy for start time and slipping rate during production.

## 1. Introduction

Large-scale digital fabrication processes have been generating an ever-growing research interest in the last few decades. They showcase potential strategies to enhance efficiency in the construction industry while offering increased flexibility in design [1]. In the case of digital fabrication with concrete, the control over early age hydration and rheology [2,3,4,5,6,7] eliminates the need for formworks or allows for alternative material-efficient formwork systems instead of bulky conventional formwork constructions [8].

This paper focuses on smart dynamic casting (SDC), a robotic slip-forming process pioneered by Dr. Ena Lloret-Fritschi [9]. In SDC, a small formwork with a maximum height of 40 cm is used to produce one-to-one scale architectural elements such as columns, mullions and folded structures [9]. The formwork is moved along a predefined path using a 6-axis robotic arm [9,10,11,12] or a linear axis [13,14,15]. The formwork is open at its top and bottom, it is continuously filled from the top and shapes the concrete leaving through the bottom along its movement (see Figure 1).

Shaping concrete in SDC requires that the concrete sets on demand and builds up strength during the robotic experiments at an expected rate based on previously conducted measurements [11,17]. First, a large retarded batch is prepared that is suitable for pumping and has a long enough open time for the whole duration of the process [17,18]. Then, the concrete is incrementally accelerated with chemical admixtures and cast into the formwork in a self-compacting state without segregation. By the time the concrete exits the moving formwork, its strength has evolved to support the weight of the material above, while maintaining its ability to deform plastically. The capacity of the material to deform at the bottom of the formwork along different paths provides the ability to produce a set of different geometries with the same formwork [10].

The formwork in SDC is related to the geometry of the produced element. The process has already proved itself suitable for non-standard columns (150 × 150 mm rectangular, d = 212 mm circular, r_1_ = 125 mm, r_2_ = 80 mm elliptical cross-section) and mullions with a changing cross-section (from 70 mm × 100 mm to 70 mm × 180 mm) [14]. Then, the SDC fabrication of thin folded structures (25 mm wall thickness and 500 mm long sides) was explored as a potentially less wasteful and less labor-intensive manufacturing method [14,19] for this material-efficient structural system with increased stiffness relative to the thickness [20]. However, after the failures of the initial empirical studies it became obvious that the experimentally defined guidelines for the slipping criterion of SDC [10] are not applicable for all formwork geometries. The increased surface area of thin folded formworks provided stricter slipping criterion, meaning a narrower process window [11].

This paper takes a material-focused approach to study the process limitations of SDC with a theoretical framework based on the process window model published in [11]. It discusses different material compositions that could provide easier processing even for thin folded elements. In the first part of the paper, the workability and the strength evolution of the different mixes are addressed and correlated to geometrical limitations for the produced elements. The second part studies the friction between the formwork walls and the concrete on a simplified experimental setup with the material compositions defined in part one in order to validate assumptions of the process model and extract parameters for further calculations. The third part discusses the theoretical model for the SDC process to analyze and consequently handle SDC production and introduces the ‘SDC number’ to formulate the process window. In doing so, various specific examples are presented, whereby values of the SDC number specifically referring to our experimental conditions are given (directly by the upper bounds of the process window). However, it should be understood that these values for the SDC number are purely illustrative. Overall optimization of SDC will involve numerous factors that can imply a large number of optimal combinations. What our model does is to help to identify these, but it does not offer any “off the shelf universal SDC number” for any arbitrary production using SDC.

## 2. Materials and Methods

### 2.1. Materials

For the experiments described in this paper two different retarded base mix formulations, a self-compacting concrete (noted ‘mix S’) and an ultra high-performance fibre reinforced concrete (UHPFRC) (noted ‘mix U’), were tested with three suitable acceleration methods. These are summarized in Table 1 and Table 2, with additional details given in the following paragraphs.

#### 2.1.1. Self-Compacting Concrete

The self-compacting concrete, mix S, had a mix design that was adjusted from previous SDC formulations [11] in order to be more compatible with the demands for the slip-forming of thin folded structures. It comprised 0–4 mm siliceous sand aggregates, a CEM I 52.5R Portland cement and silica fume (Table 1).

The admixtures, a retarder (sucrose 99.5% from Sigma-Aldrich, Buchs, Switzerland) and a superplasticizer (BASF MasterGlenium ACE 30, Basel, Switzeland) were directly added to the mixing water. Then ‘S’ was mixed with a Collomix Collomatic XM 3-900 type forced action mixer (Gaimersheim, Germany) for 7 min.

After this, the mixes were activated at specific times after initial mixing by using activators, for which two types were considered (Table 2):a set accelerator: SikaRapid C-100 (reportedly based on Ca(NO_3_)_2_·4H_2_O [21]). These mixes are noted S_C.a combination of aluminium sulfate solution (produced in house by dissolving Al_2_(SO_4_)_3_·18H_2_O in water to obtain 260 g/L concentration) and a superplasticizer (BASF MasterGlenium ACE 30). These mixes are noted S_A.

In both cases, the accelerators were added prior to casting. This was done at three different times after the initial water addition (1, 3 and 5 h), simulating the conditions for the slip-forming process. For acceleration, the concrete was mixed again with the same mixer for 5 min. The exact activation time-dependent composition is described with the third digit of the name. For example, S_C_3 indicates that the retarded S mix was accelerated with accelerator C100, three hours after the initial mixing with water.

It should also be noted that for the S_C mixes, the amount of accelerator added was increased over time, while for S_A only the superplasticizer addition increased. Many preliminary trials showed these choices to give the best performance in each case. Discussing why this is the case is beyond the scope of this paper. However, to illustrate the benefit of time-dependent dosage, the yield stress of the accelerated samples with both time-dependent and constant dosages (marked with “°” after the composition name) are compared in Section 3.1.1 with more information about the choice of time-dependent dosages. 

#### 2.1.2. UHPFRC

The UHPFRC, mix ‘U’, had a mix design based on the formulation developed by Hajiesmaeili and Denarié [22]. That original formulation was modified to enable its retardation and acceleration on demand. It contained 0.1–0.4 mm silica sand, a CEM I 52,5N Portland cement, silica fume, two types of limestone fillers and alkali-resistant Zirconia coated glass fibres (Table 1).

The admixtures (the same retarder and superplasticizer as mix S) were pre-dissolved in the water, added together to the powder parts and mixed for ten minutes with a forced action mixer. The ‘mix U’ was activated with the same self-prepared aluminum sulfate solution which was used for S_A, however, the extra superplasticizer was replaced by water. This modification was necessary to prevent the rapid skinning on the surface due to the evaporation of water [23]. The addition amounts were defined in iterative experiments to achieve similar hydration kinetics in the first hour for S_C and S_A mixes. The resulting accelerated mixes are noted U_A (Table 2).

The activator is added according to the same schedule and mixing protocol as for S_A. For U_A, the accelerator amount is decreased over time. It is important to remember that the third digit of the name again refers to the time of addition, but also relates to a modified activator dosage. For example, sample U_A_3 indicates that mix U was activated by aluminum sulfate 3 h after initial mixing and the specific dosage used in that case can be looked up in Table 2. As before, the choice of how to vary the accelerator dosage was the result of a large number of preliminary tests only reported in this work with the yield stress comparison of the time-dependent and constant acceleration (marked with “°” on Figure 3). More information about the choice of time-dependent dosages is given in Section 3.1.1.

### 2.2. Methods

#### 2.2.1. Slump Tests

The workability of all mixes, before and after acceleration, was evaluated using a Hagerman’s mini-slump cone with dimensions of 60 mm × 70 mm × 100 mm. The S and U retarded mixes were tested approximately 45 min after water addition. The accelerated mixes were tested 6 min after the start of activation in order to model the time difference between the start of acceleration and the arrival in the formwork during a slip-casting experiment. These accelerated samples were activated at 1, 3 and 5 h either with constant or with time-dependent accelerator dosages. The yield stress was calculated from the slump flow [24].

#### 2.2.2. Penetration Tests

Penetrometer measurements were performed to monitor the yield stress evolution of the accelerated samples, using protocols described in detail in another paper [11]. In short, a cylindrical needle (d = 19 mm, h = 4 mm) was repeatedly immersed in the sample at different positions with a constant speed of 1 mm/s. The peak force recorded during these measurements is related to the yield stress [25].

#### 2.2.3. Calorimetry

Isothermal calorimetry measurements were performed to determine the hydration kinetics both in the retarded and accelerated compositions, according to the protocol described in detail in [26]. Additionally, the amounts of heat released in the timespan of the penetrometer measurements were compared with the measured strength evolutions.

#### 2.2.4. Friction Tests

A friction test was designed to investigate the stresses occurring during the slipping process at the interface of the concrete and the formwork by pulling out material strips from concrete. To make it representative of our particular systems, these strips were made of the same material as the interior of the SDC formwork. Moreover, as explained below, to see the evolution of friction force over time the test is done with 5 strips, each pulled out at a different time.

The strips were 0.5 mm thick and made of hard, clear PVC sheets with a width of 50 mm and a height of 350 mm. They were distributed with equal spacing along the middle axis of a container with 60 × 430 × 310 cm inner dimensions. For better control of their position and displacement, the strips passed through a loose guiding piece providing a 1mm wide channel on the top and rest tightly in a slit at the bottom (see Figure 2). In order to avoid friction from the bottom slit during the measurement, each strip was pulled up 10mm immediately after the container was filled (see the middle of Figure 2).

The concrete was placed according to two different protocols:It was first accelerated as a whole and then cast in the box all at once.It was gradually placed using 12 smaller batches, each accelerated separately prior to placing. This procedure was done so as to raise the level of concrete by 1.8 cm every 3 min, which mimics formwork filling during the slip-forming process.

In both cases, the acceleration started 1h after water addition and the mixing time was 2 min. However, for the “one-go” filling a forced action mixer was used, while for the gradual filling an overhead stirrer had to be used because of the smaller batch sizes. With both filling protocols, the target was that the strips were 200 mm deep in concrete after leaving the bottom slit. Any deviation from that target height is noted and used for the calculations accordingly.

For each measurement, the box was positioned so that the strip to be pulled out could be clamped to the cross-bar of a Zwick universal testing machine, while the box was fastened to the platen of this device. Then, the strip was pulled out at constant 6 mm/min speed for 6 min, imitating the slipping speed of SDC used for producing non-standard thin folded prototypes [12]. For each strip series, the first test was done 105 min after water addition, while the subsequent tests (strips) were measured in 10 min intervals after that. The force needed for the displacement was logged during the pull-out of the strips. The pull-out stress was calculated from this force with the contact surface of the concrete and the strips. This test should not be confused with the standardized pull-out tests for hardened concrete where reinforcement is pulled out of hardened concrete samples. In our case, the material was still a yield stress fluid at the time of testing. This is why we define the test as a “friction test”. However, what is measured is a “pull out stress”, which might represent a source of confusion otherwise.

In parallel to the friction tests, penetration tests were conducted on companion samples. These were prepared from the same retarded base mix and were accelerated with the same compounds at the same time as the samples tested for friction.

## 3. Part 1: Mastering the Yield Stress Evolution

### 3.1. Results

#### 3.1.1. Spread Flow

Figure 3 represents the yield stress obtained by the spread flow measurements that were conducted on both retarded and accelerated samples, investigating the rheological consequences of acceleration for fluidity at the time of casting (values given in Table A1 of the Appendix A). The non-accelerated mixes, S and U mix, are shown for time 0 with open symbols. The accelerated samples are reported as a function of the time at which the accelerator was added (1, 3, 5 h).

The non-accelerated mix U has significantly larger yield stress than the S mix, despite the S mix still being fluid enough for pumping [11]. The effect of adding an accelerator is shown for both constant dosages (Figure 3, left) and time-dependent ones (Figure 3, right).

For the S mixes, it can be seen that these additions caused an initial drop in the yield stress which was increasingly less important as the addition time is increased from 1 to 5 h, both for the C and the A type accelerators. However, Figure 3 (right) shows that if the accelerator dosage is increased as defined in Table 2, the yield stress can be maintained. This is important for maintaining constant process conditions in SDC.

When changing the accelerator of the S mix to the aluminium sulfate solution (S_A*) there was flow loss (see Table A1 in Appendix A), which led us to consider additional and simultaneous additions of superplasticizer to avoid this problem. However, a dosage giving initial yield stress similar to S_C_1 leads to a great increase in yield stress at later acceleration times than what was obtained previously (S_C_1°). As before, this change could be mitigated by changing the superplasticizer content of the accelerator in relation to the time of activation (Table 2), once again providing a solution to maintain yield stress constant throughout the process (Figure 3, right).

For the U_A mixes, the yield stress increased upon acceleration, contrary to the S_A mixes. In this case, however, the yield stresses remained low over time, even with a constant accelerator dosage (U_A_1°). Due to this, there was no need to add additional amounts of superplasticizer. Additionally, to mitigate the small yield stress increase taking place at late acceleration times, small reductions of the aluminium sulfate solution proved to be most effective (Figure 3, right).

#### 3.1.2. Penetration Tests

Figure 4 shows that the evolution of penetration resistance in S_C mixes is very similar, irrespective of the time at which the accelerator is added. This, however, relates to the situation in which the accelerator dosage is adapted to maintain constant initial yield stress after its addition, as explained above (Figure 4) [11]. The initial penetration resistance is expected to be similar, since this also represents a measurement of yield stress [27]. However, the fact that after this the increase in yield stress should be similar is not self-evident. For SDC, it is, however, a very useful result because it maintained similar material evolution kinetics throughout the process. In particular, for the empirically defined slipping range of 4–6 N of penetration resistance it defined a fixed time of occurrence (50 min) as well as a fixed window of operation.

With respect to the theoretical model presented later in this paper, we can point out that when the data are plotted on a log-log scale, their appearance as a straight line indicates that the yield stress is increasing as a power law of the time elapsed after addition of the accelerator.

A consistent yield stress evolution over the 5 h timespan was also achieved by the aluminate-based accelerator (S_A series), provided by the concentrated aluminium sulfate solution combined with a superplasticizer (see Figure 5). In this case, it required the superplasticizer dosage to be varied. As before, the fact that similar initial yield stresses were obtained is not surprising, since dosages were adjusted to obtain similar flow spreads. The fact that the yield stress growth over time should be the same is not trivial, but it is, as previously mentioned, very useful.

As can best be seen in the log-log plot (Figure 5, right), this acceleration method also leads to power-law increases in yield stress over time. In this case, however, it provides lower exponents (lower slopes on the log-log plot). Apart from this, we observed that the late measurements showed larger scatter than for the S_C mixes. The experimental slipping range was reached again around 50 min.

The U_A mix also shows similar penetration resistance evolutions after acceleration addition, irrespective of the addition time (Figure 6). Similarly, as in the two previous cases, this was obtained by adjusting the accelerator dosage over time to get the same initial yield stress. However, in contrast to the two previous cases, the U_A mixes required the dosage to be decreased rather than increased. The reasons for these differences are due to the underlying mechanism of each acceleration method and are beyond the scope of this paper, which focusses instead on showing how consistent behaviour can be obtained in order to enable a more robust SDC process.

Importantly, for the process modelling, we once again found that a power-law can describe the strength evolution well after accelerator addition (Figure 6, right). The exponents of the strength evolution do not depend on the time of accelerator addition and are in a similar range to the S_C samples. However, the target value of 4–6N of penetration resistance was reached slightly earlier than before.

#### 3.1.3. Calorimetry

As shown in Figure 7, the non-accelerated S mix had a particularly long induction period, with an onset of the main hydration at around 7–10 h. This time frame provides workability retention for a whole work-day that is needed for our typical SDC production conditions. The figure also shows a strong reduction in the duration of the induction period when accelerator C was added. However, while the initial hours are very similar, the hydration kinetics differ after about 4 h. The shape of the three curves also suggests that there is an inversion of the aluminate and silicate hydration, with the most intense peak probably reflecting sulfate depletion and the main silicate peak following at around 15–20 h depending on the time of accelerator addition [5]. This implies that the main strength gain is still rather delayed, which is something to consider when producing tall structures.

This pattern is not seen in the S_A samples (Figure 8). The shape of their hydration curves is very similar and does not suggest an inversion of the aluminate and silicate peaks. The silicate peak appears to occur around 10 h, quickly followed by the sulfate depletion peak. Apart from this, we also observed a much lower heat flow and thus a much lower reaction rate in the induction period. In fact, the values are similar to those of the non-accelerated S formulation.

Importantly for SDC process control, the onset of hydration occurs earlier with this acceleration method than for the S_C mixes and is also independent of addition time.

Looking next at the U_A samples, it is evident that the induction periods are shorter and the heat flows are intermediate between S_C and S_A (Figure 9). The hydration curves do not overlap as well as S_A, with onset times occurring later for earlier additions. Finally, there is no peak inversion between aluminates and silicates.

### 3.2. Discussion

#### 3.2.1. Initial Yield Stress after Accelerator Addition

To begin this discussion, let us recall that we are considering a retarded concrete batch being pumped and activated before being delivered into an SDC mould that is slipped vertically. For this process to run smoothly throughout the production of an element, we would like the residence time of the concrete in the formwork to be constant. This means that the time needed to reach the target “exit-yield stress” should be constant.

The results show that this is only possible if the accelerator dosage is varied over time. The concrete is already ageing before it gets pumped and as a result even small amounts of hydration products formed in this period can influence rheological behaviour substantially because they create a high specific surface that has a profound consequence on yield stress [28,29].

For the S_A series, the addition of an accelerator initially reduces the yield stress with respect to the base concrete. For both accelerators, this may be partially due to the water they contain. Further, the presence of calcium ions can additionally promote superplasticizer adsorption. For the A accelerator, the additional amount of superplasticizer plays an important role. Thus, in both cases, we can expect that the accelerator addition initially increases the amount of adsorbed superplasticizers. In any case, the fact that the superplasticizer addition decreases the yield stress of the base concrete also shows that increasing its dosage can counteract the yield stress increase obtained for constant dosages in the accelerated compositions.

For the U series, the accelerator addition actually increases the yield stress. This shows that lower dosages should be used at later addition times, or else the yield stress is higher than for early hydration times. The reason the accelerator addition increases yield stress in the U mix while decreasing it in the S mix is that the A accelerator does not include any superplasticizer in this case. Therefore, the addition of the aluminium sulfate is creating additional surfaces (most likely ettringite) that consume the superplasticizer. Thus, the increase in the effective volume fraction causes yield stress to increase.

#### 3.2.2. Penetration Resistance/Strength Evolution

After ensuring that we could have fixed yield stress after acceleration, and a consistent evolution during slipping, we are interested in knowing the rate of evolution of this yield stress. There are two reasons for this. First, as the yield stress increases with time after acceleration, there is a distribution of yield stresses in the formwork. This means that the total force needed to pull up the formwork must be determined from the integration of yield stress from the bottom to the top of the formwork. The integration can be done over the known residence time of concrete in the formwork: the oldest concrete being at the bottom and the youngest at the top (more information is given in Section 4.1). The second reason for our interest in the rate of yield stress growth is that it can tell us if the concrete continues to gain enough strength to carry the additional weight of the object being produced.

We gain insight into these questions by examining penetration resistance measurements, which provide crucially important information for SDC as well as for other digital fabrication processes [30]. Our results mainly show that provided the post-acceleration yield stress is kept constant (as described in the previous section) the yield stress evolution is very similar regardless of the time of acceleration. We also observed that the rate of yield stress change after acceleration can be quite different between the different cases. Importantly, we see that the mixes S_C and U_A provide the least scatter and therefore offer very appealing solutions for a smooth production control by SDC.

In the case of the C accelerator, it should be noted that the manufacturer defines maximum dosages that ought not to be exceeded. This puts a limit on the rate of strength gain and therefore also on that of SDC production. As an example, for 5 h production time, with a 30 cm high formwork and a mix that requires 60 min to reach the process window, the slipping speed can be calculated as 5 mm/min. This allows for a maximum 150 cm slipping path during the experiment, instead of the desired 300 cm that would allow for more meaningful construction scale components.

As the A accelerator is a solution containing a significant amount of water, the theoretical upper limit of the dosage leads to the highest possible w/b ratio allowed by the building code. By increasing the concentration of the accelerator to a stable suspension, full-scale components could be produced with several meters per hour of vertical building rate in reasonable times [26].

#### 3.2.3. Cement Hydration

The calorimetry results of the accelerated samples in the first hour all show an increased heat release that we can relate to rapid precipitation of hydration products following the addition of accelerator (more so with aluminium sulfate). This does not directly translate to yield stress increase because, as noted above, in the S concrete compositions additional water and superplasticizer adsorption is expected.

As already noted in Section 3.1.3, all three S_C samples (Figure 7) show signs of sulfate depletion [31]. This means that the silicate hydration remains quite delayed and that the sample will have an intermediate strength for an extended period of time. This may limit the maximum stable height of SDC objects. Excessively delayed silicate hydration probably also makes the finished elements produced with the S_C mix more susceptible to drying shrinkage. During production and curing afterwards without formwork may lead to loss of water before most of the C3S has hydrated.

In contrast, the aluminium sulfate type of acceleration, either with the ‘S mix’ (Figure 8) or with the ‘U mix’ (Figure A1), shows no flash set or sulfate depletion. One reason is that this type of accelerator contains aluminium and sulfates in the ratio needed to produce ettringite. The time when the main silicate peak (S_A) or the merged main hydration peak (U_A) occurs largely depends on the accelerator dosage and it is discussed in another paper [26].

## 4. Part 2: Friction Tests

### 4.1. Results

In what follows, the results of the friction tests are reported as pull-out stress versus displacement. The pull-out stress is calculated using the contact surface of the concrete with the immersed strip at each point of the measurement.

Results for S_C, S_A and U_A with an accelerator addition 1 h after initial mixing are reported in Figure 10. The figure on the left shows the situation in which the box is filled in one go by the concrete, while for the figure on the right the concrete was added gradually in small layers prepared individually one after the other to mimic the continuous SDC process.

Figure 10 shows that the values for direct casting are higher than for progressive addition. This is because in the latter case, less time has elapsed since acceleration in the upper layers than for the concrete in the base than when the concrete is filled all at once. Apart from these differences, we note that in all cases the stresses are initially higher and then decrease towards a plateau. Thus, each of these tests are characterised by a peak stress and a plateau stress taken as the average of stress for displacements between 15 and 20 mm, when these stresses are found to be roughly constant.

Interestingly, the ranking of plateau stresses is not the same depending on the filling mode (see Figure 10). For the direct filling, the highest peak and plateau stresses are found for S_C, followed by U_A and then S_A. For the progressive filling, the highest stresses are in U_A. However, its peak is lower than that of the S_A.

To determine the relation of these friction measurements to yield stress, we first report them with respect to the penetration resistance determined for the material with the same activation time as the one present at the bottom of the box. This is done for the peak and the plateau pull-out stresses respectively in the left and right parts of Figure 11. Independently of the concrete type or the filling mode, both data series can be well fitted with a line forced through the origin. This is important for our paper since it establishes the possibility that the friction force, represented here by the pull-out force, is proportional to the yield stress found at the bottom of the SDC formwork.

To more directly relate the friction measurements to yield stress, and to account for the yield stress variation throughout the gradually placed samples, we computed average yield stress from penetrometer measurements done on samples representing the concrete at the top and bottom of the box $\tau_{0,h7}\;and\;\tau_{0,h1}$ in Figure 12.

Plotting the peak and plateau pull-out stress versus the average yield stress, we again see a linear relation (see Figure 13). However, the rate should depend not only on the material composition but also on the filling method. For the gradual filling, the average penetration resistance was obtained by measuring the penetration resistance of the material equivalent to both the first and the last batch filled into the box. For the same composition, the friction is 2–3 times lower when the container is filled gradually compared to filling it at once.

### 4.2. Discussion

During SDC, the upper limit of the process window is related to the friction that results from the sum of the spatially variable shear stress along the surface of the formwork. The friction tests presented in this paper were designed to determine friction laws that may be used to describe this process as well as the parameters that they contain.

In our experiment, the surface in contact with the formwork was expected to be a cement paste, which allows the formation of a lubrication layer along the wall. This explains why the pull-out stresses that we measure are substantially lower than the average yield stress of the concrete. Values are in fact roughly an order of magnitude lower, which corresponds to the range of yield stresses of the cement paste.

Our results show that the friction force depends more or less linearly on the yield stress and that the frictional coefficients depend on the mix design, accelerator and filling mode. Qualitatively, it is understood that the gradual filling mode reduces the overall friction force due to the gradient of material age in the box with lower average age than with simple filling. However, the fact that the filling mode inverses the ranking of friction forces is less trivial. This result implies that the rate of structural build-up plays an important role in this process, and that SDC may only be reliably simulated when this is accounted for. Specifically, it is important to obtain trustworthy measurements of both the friction coefficient and the structural build-up kinetic law.

All friction experiments (Figure 10) show a peak stress at the start of the pull-out with a higher resistance than later when a plateau stress is ultimately reached. Consequently, the initiation of slipping is substantially more difficult than its continuation, thus intermediate pauses in slipping should be avoided, as had already been recognized in the Ph.D. Thesis of Lloret [9]. Results from this work underline that this comes from a difference between static and dynamic friction. As both increase with hydration, the need to avoid stop-go is not caused by hydration but is in the very nature of the difference between both those friction modes. Our results also express this by demonstrating the process window for SDC, this implies that the process window at the start of slipping is narrower than it is for the running process.

The underlying reason for the higher initial stress is a structural breakdown of the interface material [32]. In partial support of this argument, it is observed that the factor between the initial peak pull-out stresses and the plateau stresses are similar in all tests: 3.6–4.8 for direct filling and 2.2–3.0 for gradual filling (see Table A2 in Appendix A). The fact that the stress reaches a plateau implies that the interfacial material reaches a steady state of structural breakdown at some point.

## 5. Part 3: A Theoretical Model for the SDC Process Window

### 5.1. Reformulating the Process Window with the SDC Number’

The slip-forming process model is based on the force balance in the formwork during slipping (see Figure 14). SDC can be understood as gravitational extrusion through a formwork-size nozzle where the weight of the concrete in the formwork is the driving force that has to overcome the friction acting along the formwork walls. Thus, the ratio of the surface and the volume in the formwork plays a significant role in successful slipping and will be referred to as hydrodynamic radius $r_{hy}=\frac{2\cdot Volume}{Surface}$.

For the dimensions used for different SDC cases, we find the following values of $r_{hy}$: Column with a square cross-section of 150 mm side length: $\;r_{hy}=75\;mm$ [9];Column with a circular cross-section of 212 mm diameter: $\;r_{hy}=106\;mm$ [9,10,13];Column with an elliptical cross-section of 125 mm and 80 mm radii: $\;r_{hy}=96\;mm$ [9];Façade mullion with rectangular cross section of 100 mm and 70 mm: $r_{hy}=41\;mm$ [14];Folded structures with 25 mm thin 500 mm long sides: $\;r_{hy}=25\;mm$ [11].

The vertical stress at the extrusion point can be expressed as the difference between the hydrostatic pressure ρgH of concrete in the formwork, expressed with the formwork of height H, and the frictional stress $\tau_{Fr}$ acting along the formwork walls (see Figure 14). As $\tau_{Fr}$ varies along with the formwork height, we write:(1)$$\sigma_{VE}=\rho gH-\frac{2}{r_{hy}}\int_{0}^{H}\tau_{Fr}\left(z\right)dz.$$

This process is constrained by two failure types: the tear-off and the flow-out [10]. First, concerning the tear-off, the vertical stress $\sigma_{VE}$ has to be positive. Second, to avoid flow-out, the Tesca criterion must be satisfied, which implies that $\sigma_{VE}$ must be lower or equal to twice the yield stress of the concrete leaving the formwork $\tau_{0}\left(t_{Extr}\right)$. Both of these conditions can be summarized as:(2)$$0\leq \sigma_{VE}/2\leq \tau_{0}\left(t_{Extr}\right).$$

From Section 3 and in particular Figure 4, Figure 5 and Figure 6, the yield stress can be considered to evolve as a power-law, scaling to the time *t* after acceleration. Further, from Section 4 and in particular Figure 12, the friction stress can be considered as proportional to the yield stress:(3)$$\tau_{0}\left(t\right)=\alpha_{C}t^{\beta_{C}}$$
(4)$$\tau_{Fr}\left(t\right)=\alpha_{Fr}\;\tau_{0}\left(t\right)=\alpha_{Fr}\alpha_{C}t^{\beta_{C}}.$$
where $\alpha_{Fr}$ is a friction coefficient, $\alpha_{C}$ is the prefactor of the power law for yield stress effective and $\beta_{C}$ is its exponent.

For a constant vertical building rate *V*, we can rewrite this in relation to any position *z* along with the formwork height:(5)$$\tau_{Fr}\left(z\right)=\alpha_{Fr}\alpha_{C}{\left(\frac{z}{V}\right)}^{\beta_{C}}.$$

Substituting this into (1) we get:(6)$$\sigma_{VE}=\rho gH-\frac{2}{r_{hy}}\int_{0}^{H}\alpha_{Fr}\alpha_{C}{\left(\frac{z}{V}\right)}^{\beta_{C}}dz=\rho gH-\frac{2\;\alpha_{Fr}\alpha_{C}}{r_{hy}\left(1+\beta_{C}\right)V^{\beta_{C}}}H^{\beta_{C}+1}.$$

Equation (5) can also be used to write conditions at the level of extrusion. There, the paste is the oldest in the formwork and the yield stress at the time of extrusion $\;\tau_{0,\;Extr}$ can be written as:(7)$$\;\tau_{0,\;Extr}=\alpha_{C}{\left(\frac{H}{V}\right)}^{\beta_{C}},$$
substituting this equation into (6) gives:(8)$$\sigma_{VE}=\rho gH-\frac{2\;\alpha_{Fr}\;\tau_{0,\;Extr}}{r_{hy}\left(1+\beta_{C}\right)}H$$
which in turn can be substituted into the expression for the operating window given in (*2*):(9)$$\frac{H\alpha_{Fr}}{r_{hy}\left(1+\beta_{C}\right)}\leq \frac{\rho gH}{2\;\tau_{0,Extr}}\leq 1+\frac{H\alpha_{Fr}}{r_{hy}\left(1+\beta_{C}\right)}.$$

As proposed by Szabo et al [11], this process window can be more conveniently formulated with a dimensionless parameter ξ given by $\frac{H\alpha_{Fr}}{r_{hy}\left(1+\beta_{C}\right)}$, giving:(10)$$\xi \leq \frac{\rho gH}{2\;\tau_{0,Extr}}\leq 1+\xi .$$

In this paper, we use a slightly different notation, by introducing a dimensionless yield stress:(11)$$T_{0,Extr}=\frac{\tau_{0,Extr}}{\rho gH},$$
and a dimensionless number that we call the SDC number, given by:(12)$$SDC=\frac{1}{2\;\xi }.$$

The meaning of this number is best understood by rewriting it as:(13)$$SDC=\frac{\left(1+\beta_{C}\right)}{\frac{H}{r_{hy}}\alpha_{Fr}}.$$

By doing this, the numerator depends only on strength gain, which is captured by the exponent of the yield stress growth $\beta_{C}$. It can be understood to represent a characteristic cohesion gain. In the denominator, the term $\frac{H}{r_{hy}}$ represents the ratio between the height of fresh concrete over a characteristic thickness of the formwork section. It can be thought of as a friction equivalent of a slenderness ratio, whereby high values of the ratio lead to high friction and vice versa. Together with the friction coefficient $\alpha_{Fr}$, this implies that the denominator relates to a characteristic friction of the system. All in all, it implies that the dimensionless SDC number expresses a ratio between a characteristic cohesion and a characteristic friction.

With this in mind, we are now in a better position to discuss the process window that we rewrite as:(14)$$\frac{SDC}{2\;SDC+1}\leq T_{0,Extr}\leq SDC.$$

This indicates that for the process to work, the dimensionless yield stress should be lower than the SDC number. Otherwise, the material is too cohesive, friction is too high and there will be a pull-off. It also indicates that to avoid flow out, the dimensionless yield stress should be larger than $\frac{SDC}{2\;SDC+1}$. For large SDC numbers, where cohesion gain is fast and/or friction is low, the dimensionless yield stress should be larger than ½ (meaning $\tau_{0,Extr}>\frac{\rho gH}{2}$ based on Equation (11)).

With respect to the main objective of this paper, which is to examine the feasibility of producing thin folded elements by SDC, the effect of a change in hydrodynamic radius $R_{hy}$ can be quantified if the other parameters remain unchanged:(15)$$R_{r_{hy}}=\frac{SDC_{1}}{SDC_{2}}=\frac{r_{1}}{r_{2}}.$$

In the case of formworks for thin folded geometries, the increase in surface area compared to the change in volume results in a hydrodynamic radius three times smaller than the one previously used in SDC formworks for columns. Thus, the process window for slipping is three times narrower for thin folded structures than for columns. With the narrow process window for thin folded forms, it is therefore even more important to have consistent yield stress evolution over time than it is for columns.

### 5.2. Practical Considerations

Friction force measurements from Section 4 and Equation (4) make it possible to determine $\alpha_{Fr}$. However, we should recall that the friction is substantially higher at the start of slipping and then decreases. Thus, we describe each material sample with two $\alpha_{Fr}$ parameters, reported in Figure 15 along with their standard deviation. In this regard, we note that the $\alpha_{Fr}$ parameters show only a limited variability within a given data series, and this most likely originates from limitations in the friction measurement setup.

With such estimates of $\alpha_{Fr}$, the boundaries of the process window for each material can be determined. To illustrate this, we consider a typical formwork for thin folded geometries with $r_{hy}=0.025m$ and H=0.3m. Moreover, we use the values of $\beta_{C}$ determined for our different mixes (see Table A3 in the Appendix A). First, we determine the dimensionless yield stress from Equation (11). The operation window is then expressed in the series of plots showing the dimensionless yield stress between the ratio $\frac{SDC}{2\;SDC+1}$ and *SDC*. Values are reported in Figure 16 with the example of ‘S_C_1’ and ‘U_A_1 gradual’ for both values of $\alpha_{Fr}$ (the plots with the other mixes and filling methods can be found in the Appendix A in Figure A2, Figure A3, Figure A4 and Figure A5). The operation window is shown to be narrower at the start and wider during slipping for all material and filling types. Further, S_C_1 mixes filled either all at once or gradually provide the tightest process window of the three compositions, which makes it the least favourable mix design for producing thin folded structures. U_A_1 filled gradually provides the best conditions for formworks for thin folded structures based on the width of its process window for the start-up and continuous slipping.

It can be observed that a minimum waiting time is needed before the start of slipping to avoid collapse, for example 55 min for ‘U_A_1 gradual’ (see Figure 16, right). A minimum waiting time also exists for S_C_1, but cannot be seen in the measured data. From Figure 16 (left) it can, however, be inferred that this value would lie around or slightly below 45 min. For this mix, we can additionally see in the same figure that the slipping should not start later than 70 min. For mix U_A_1, an upper waiting time must also exist but is not observable in the data reported in Figure 16 (right). Concerning mix S_C_1, it is important to note that the slipping initiation has a relatively narrow operation window. However, once the slip-forming process has successfully begun, it should continue with fewer problems due to the larger process window that is obtained after initiation owing to lower dynamic than static friction.

A previous study [13] found $\alpha_{Fr}=0.12$ determined with friction measurements along a rectangular formwork during continuous slipping with a similar mix composition as ‘S_C_1 gradual’ assuming $\beta_{C}\;$= 2, which is a typical yield stress evolution in such mixes. Compared to that report, here $\beta_{C}$ is determined from penetration tests and $\alpha_{Fr}$ from friction tests where the integrated friction force neglects confinement effects including thin walls or corners. Therefore, in this study $\alpha_{Fr}$ is significantly lower, which causes the possibility of overestimating the upper bounds of the process window for both the start and the slipping. The difference between these measurements emphasizes the effect of formwork geometry. The process window could be refined numerically, however, it would require separate computations for each different case. The confinement effects could also be considered by determining the upper bound of the operation window for slipping as the same as that at the start. This conservative approach provides a safety coefficient of about 3, which should largely compensate for the higher friction in confined spaces. The effect of different formwork materials would require further investigation to study the interaction between the fresh concrete and the pull-out strip.

In Figure 17, the effect of formwork geometry with different $r_{hy}$ shows a larger process window for columns with larger $r_{hy}$ than for thin folded structures with smaller $r_{hy}$. The slipping strategy can be defined based on the narrowness of the process window. For folded structures (Figure 17, Left), the slipping should start close to the lower bound of the operation window, because this can be well defined. Indeed, the lower bound only implies material failure and flow out, so its definition is subject to less uncertainty. By starting the system under such conditions, we make sure to keep it away from “pull-off” failure due to friction (upper bound of the operation window). The “pull-off” regime for thin folded structures not only is uncertain but also starts very low, thus, the risk of going into it is higher than in the case of columns.

As an example, in Figure 17 (left), the starting point for slipping folded structures is shown with a triangle at 55 min marking the time interval the concrete needs to spend in the formwork before it slips out. Provided the slipping is successfully initiated, the slipping rate could then be reduced and the acceleration time increased. In the same figure, this is indicated by the square at 65 min, with the arrow highlighting the shifting of the operation condition. Indeed, it can also be seen on Figure 17 left that once the slipping has started the “pull off” limit (upper bound) goes to much higher levels. Therefore, it becomes much safer to reduce the slipping speed, raising the operation conditions more distinctly above the lower bound and thus reducing the risk of material failure (flow out). There is, however, a balance to find here since too much speed reduction will have a negative impact on production rates.

In contrast, in Figure 17 (right), for elements with much larger hydrodynamic radii, such as column, the opposite procedure is probably best. Indeed, in this case the upper bound of the operation window for slipping initiation is much higher. It is, therefore, safe to wait longer and stay away from the lower bound. In Figure 17 (right) this is shown by the triangle at 75 min. In this case, once the slipping has started it is advantageous to move faster, reducing the acceleration time. This should, however, be done without moving too close to the lower bound. In Figure 17 (left) this is shown by the square representing the same steady-state operation condition at 65 min (as in the previous case). The arrow once again underlines the change in operation conditions, which in this case is advantageous because it increases the production speed without compromising material stability too much.

Significantly, the most suitable mix for producing a desired geometry with a formwork of a specific $r_{hy}$ is based not only on the extents of the process window, but also on the later strength evolution in the section below the formwork. Otherwise, material failure or buckling may occur [2,4,6]. Results presented in the calorimetry Section 3.1.3 show that the aluminate sulfate solution does not cause sulfate depletion (the main hydration peaks are not inverted), and therefore represents a good option to avoid unexpected delays of the tricalcium silicate hydration present in cement.

## 6. Conclusions

For thin folded geometries produced with smart dynamic casting, consistent strength evolution and extended time during which the material meets the requirements of the narrow process window are necessary. This work shows that these goals can be achieved with different set-on-demand material systems with different base mixes and accelerators. By exchanging the initially used C100 accelerator with aluminum sulfate, the limitations for dosage could be overcome due to the different mode of action. Aluminum sulfate and its working mechanism are known in the concrete industry, as it is used as one of the main components in various shotcrete accelerators [33]. However, its potential for digital fabrication could be represented best as a set-on-demand system with controlled yield stress evolution and a long open time that allows large-scale architectural elements to be produced.

Quantifying friction here provided another approach to assess different material systems. This work shows that friction is proportional to yield stress, with a coefficient of proportionality that depends on material composition and filling history. As friction is the highest in the beginning, continuous slipping is safer than start-up. The theoretical framework suggests that materials with a high exponent for strength evolution and a low frictional coefficient are beneficial. However, it also shows that all the material systems investigated have advantages and drawbacks and must be selected according to the component height and formwork hydrodynamic radius of the individual case.

As a result of this work, we propose a revised formulation of the operation window for SDC in which a dimensionless SDC number is introduced. The dimensionless yield stress at the exit of the formwork should be lower than this SDC number and larger than the ratio $\frac{SDC}{2\;SDC+1}$, which for thin folded structure has a limit value of ½. The model presented slightly overestimates the upper bound as it neglects confinement effects. This, together with the already rather narrow operation windows, underscores the sensitivity of SDC for producing thin folded geometries. It further emphasizes the importance of proper material design to achieve consistent yield stress evolution. The model presented in this paper offers a first approximation and the friction tests proposed help to define and test material requirements. More refined solutions will require either large scale SDC production testing, computational simulations, or both.

## Figures and Tables

**Figure 1 materials-13-02084-f001:**
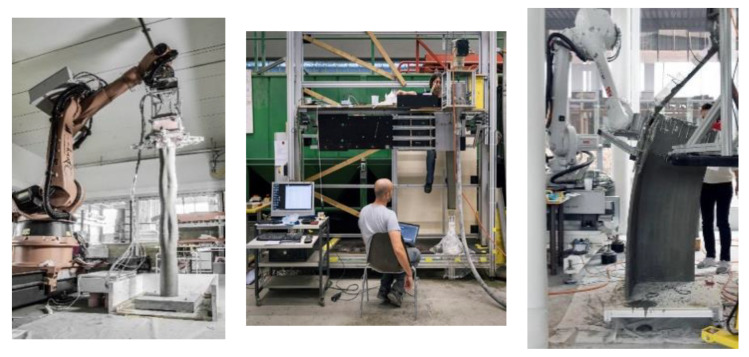
The different fabrication setups of smart dynamic casting (SDC) for producing columns on the left with a KUKA robotic arm [9], mullions in the middle with a linear axis [16], and thin folded structures on the right with an ABB robotic arm [12].

**Figure 2 materials-13-02084-f002:**
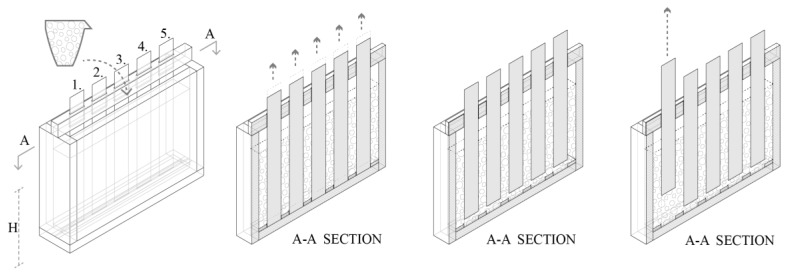
Left: friction test box is filled with concrete. Middle left and right: section of the friction tests box showing the guiding piece on top, the slit in the bottom and the final position of the strips before testing. Right: friction test performed on the first strip.

**Figure 3 materials-13-02084-f003:**
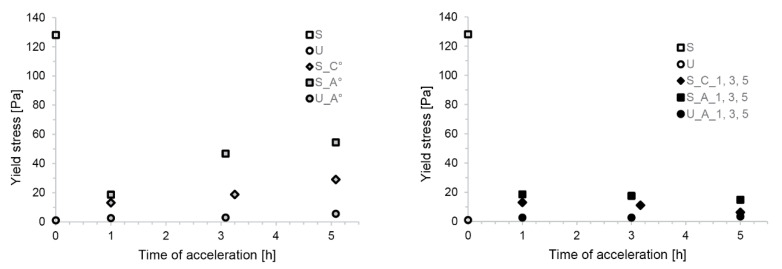
Mini-slump test results expressed as yield stress. On the left, the accelerator dosages were constant over time. On the right, time-dependent accelerator amounts were used.

**Figure 4 materials-13-02084-f004:**
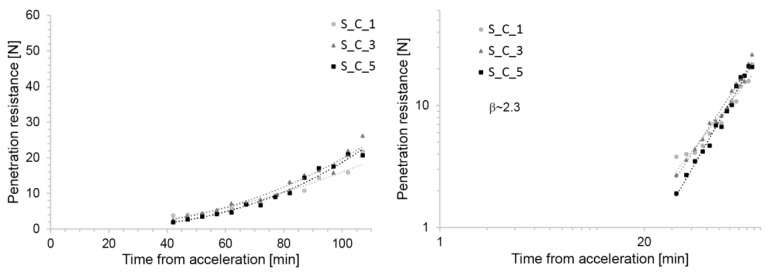
Strength evolution of the S_C Mix with C100 accelerator with increased dosage over time. (**Left**) Lin-lin scale reproduced from [11]. (**Right**) Log-log scale. The straight line obtained on the log-log scale highlights the fact that the yield stress evolution can well be fitted by a power-law (see Equation (3)).

**Figure 5 materials-13-02084-f005:**
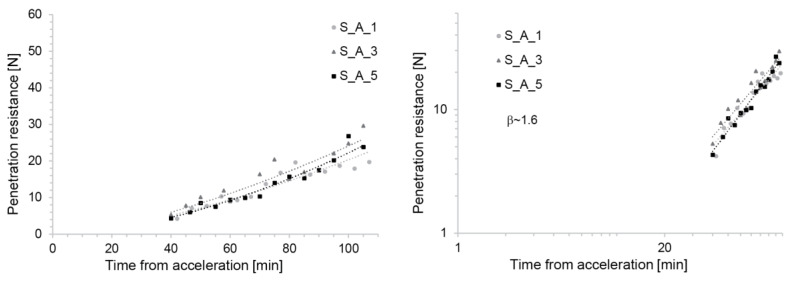
Strength evolution of the S_A Mix with aluminium sulfate solution as accelerator and simultaneous superplasticizer addition of an increased amount over time. (**Left**) lin-lin scale. (**Right**) log-log scale. The straight line obtained on the log-log scale highlights the fact that the yield stress evolution can well be fitted by a power-law (see Equation (3)).

**Figure 6 materials-13-02084-f006:**
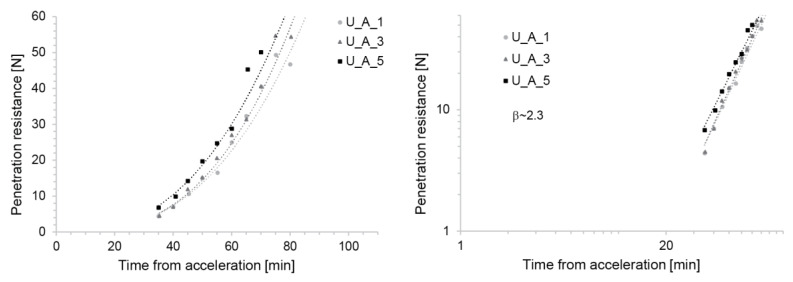
Strength evolution of the U_A Mix with aluminium sulfate solution as an accelerator with a decreased amount over time. (**Left**) Lin-lin scale. (**Right**) Log-log scale. The straight line obtained on the log-log scale highlights the fact that the yield stress evolution can well be fitted by a power-law (see Equation (3)).

**Figure 7 materials-13-02084-f007:**
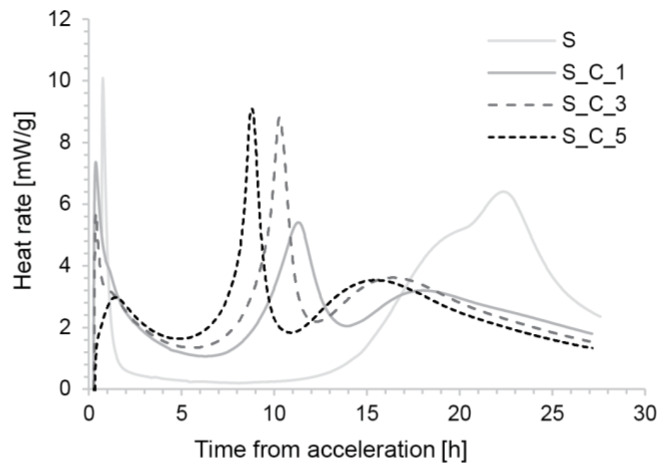
Hydration kinetics of the S mix accelerated with Sika Rapid C100.

**Figure 8 materials-13-02084-f008:**
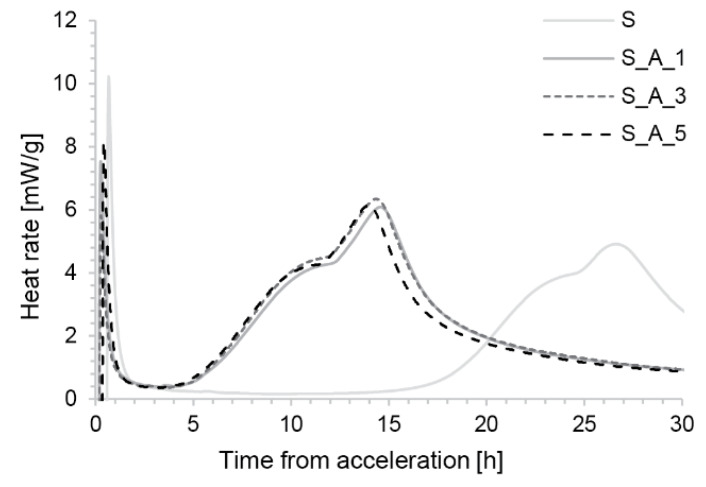
Hydration kinetics of the S mix accelerated with the combination of aluminium sulfate solution and a superplasticizer.

**Figure 9 materials-13-02084-f009:**
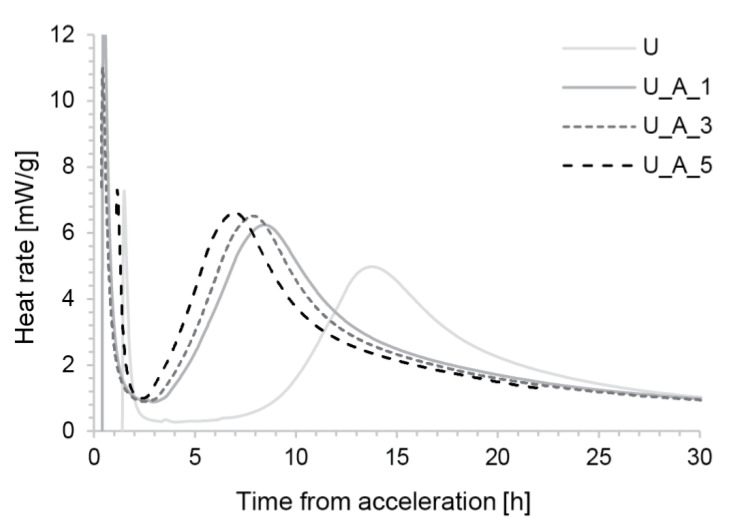
Heat flow over time from time of acceleration for the U mix accelerated with diluted aluminium sulfate solution.

**Figure 10 materials-13-02084-f010:**
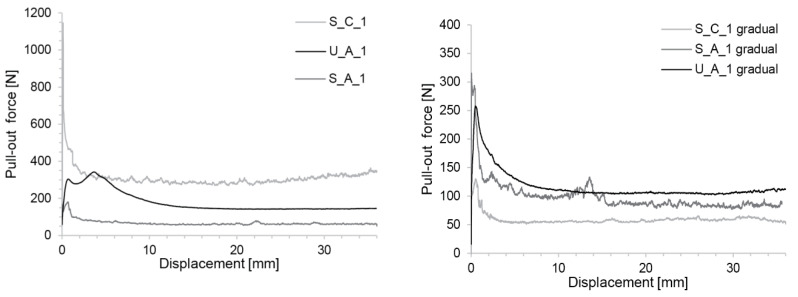
Pull-out stress of the different compositions recorded in friction tests. (**Left**) Concrete is cast in one go. (**Right**) Gradual filling mimicking SDC process.

**Figure 11 materials-13-02084-f011:**
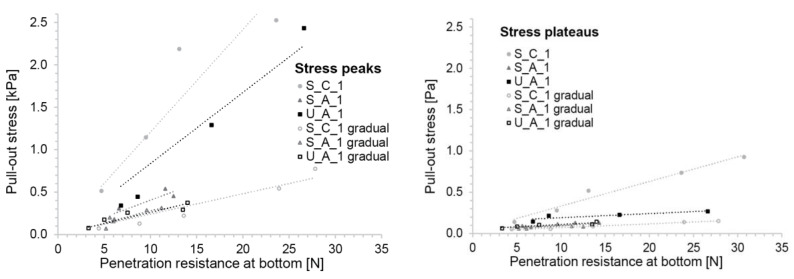
Relation of pull-out stresses to penetration resistance of the material at the bottom of the friction box. (**Left**) Peak pull-out stress, (**right**) plateau pull-out stress. Both graphs show the three concrete compositions and the two filling modes.

**Figure 12 materials-13-02084-f012:**
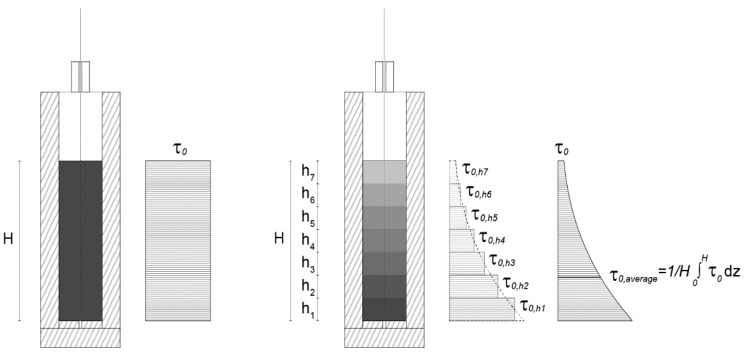
Yield stress of the concrete filled into the container at once (**left**) and gradually (**right**).

**Figure 13 materials-13-02084-f013:**
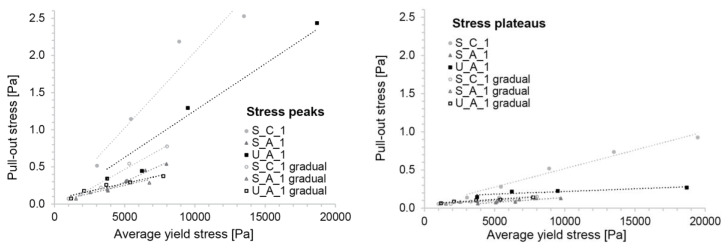
Relation of pull-out stresses to average yield stress measured by the penetrometer. (**Left**) peak pull-out stress, (**right**) plateau pull-out stress. Both graphs show the three concrete compositions and the two filling modes.

**Figure 14 materials-13-02084-f014:**
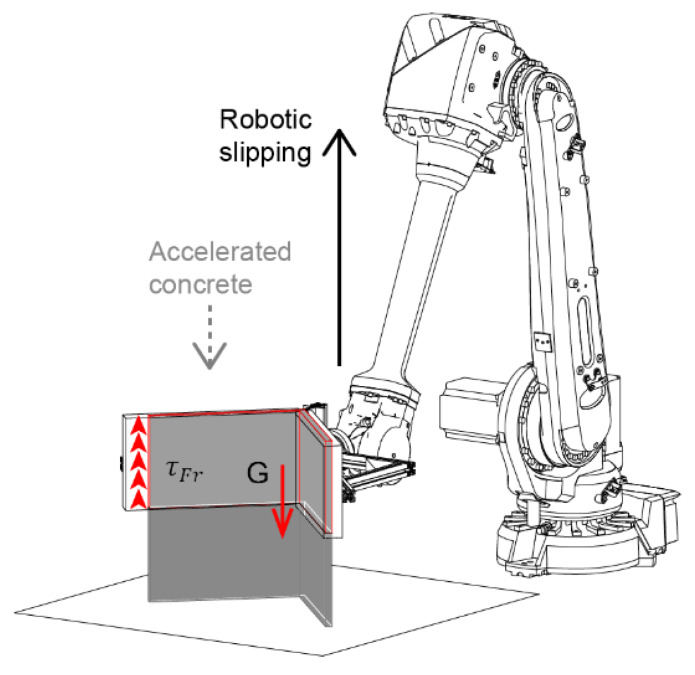
Force balance during slipping with a formwork for producing thin folded structures [11,12].

**Figure 15 materials-13-02084-f015:**
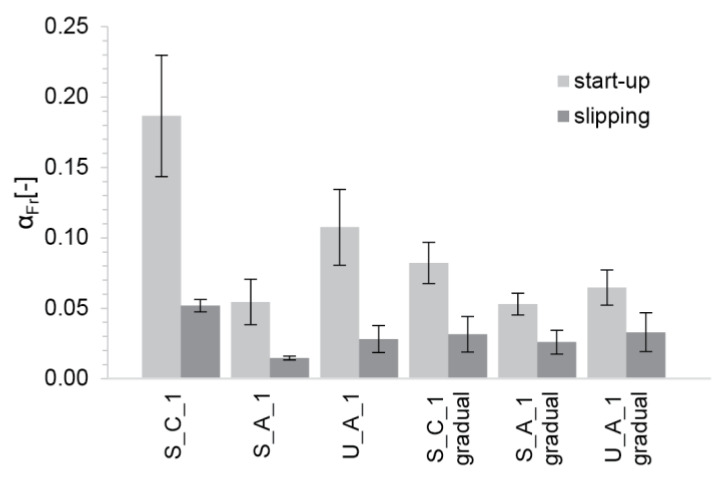
Frictional parameter calculated for each mix and filling method at the start and during slipping. The error bars represent the standard deviation of the $\alpha_{Fr}$ parameters measured over time on the different friction strips.

**Figure 16 materials-13-02084-f016:**
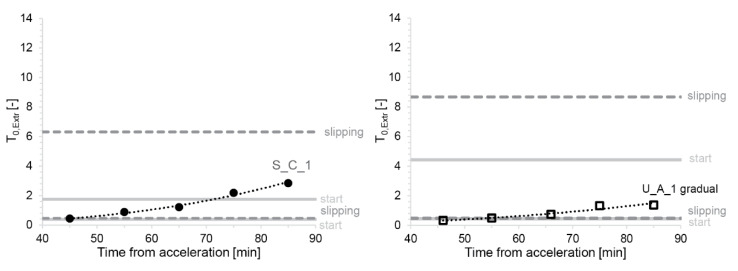
Comparison of measured extrusion yield stress shown with full circles for ‘S_C_1’ material (**left**) and with empty squares for ‘U_A_1 gradual’ material (**right**), along with the operation window between $\frac{SDC}{2\;SDC+1}$ and SDC with a formwork for thin folded structures (continuous horizontal lines indicate the start-up and dashed the slipping).

**Figure 17 materials-13-02084-f017:**
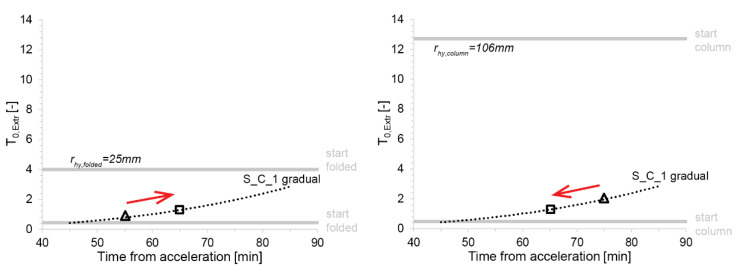
The process window between $\frac{SDC}{2\;SDC+1}$ and SDC for the start of slipping with a typical formwork for thin folded structures (**left**) and with a cylindrical formwork to produce columns (**right**). The red arrows highlight the slipping strategy for the start time and increased or decreased slipping time for later.

**Table 1 materials-13-02084-t001:** Mix designs used in mix S and U prior to acceleration.

Retarded Mix [kg/m^3^]	Mix S	Mix U
Sand (0–4 mm)	1367.60	
Quartz sand (0.125–0.5 mm)		616.40
Glass fibres (Fibre Technologies AR2500H103)		7.35
Cement (CEM I 52.5R or N)	615.06 (R)	547.50 (N)
Silica fume (BASF Masterlife SF100)	32.37	191.60
Limestone 5–15 µm d50% (OMYA Betocarb-HP)		183.10
Limestone 3 µm d50% (OMYA Betoflow-D)		419.10
Water	247.19	192.20
Superplasticizer (BASF MasterGlenium ACE-30)	1.55	6.00
Sucrose (Sigma Aldrich 99.5% purity D(+))	0.68	1.33
Ca(NO_3_)_2_ (SigmaAldrich)		0.03
w/b (-)	0.38	0.26

**Table 2 materials-13-02084-t002:** Composition of the accelerated S_C, S_A and U_A mixes and the w/b ratio of these mixes after acceleration.

Accelerator (g/l_retarded concrete_)	Mix S_C			Mix S_A				Mix U_A		
	S_C_1	S_C_3	S_C_5	S_A *	S_A_1	S_A_3	S_A_5	U_A_1	U_A_3	U_A_5
Sika Rapid C100	15.38	20.00	24.60							
Al_2_(SO_4_)_3_·18H_2_O				9.23	9.23	9.23	9.23	13.69	12.59	11.50
Water				31.95	31.95	32.20	32.44	75.48	71.85	68.21
Superplasticizer *					1.85	2.20	2.55			
w/b acc. (-)	0.39	0.40	0.40	0.43	0.43	0.43	0.43	0.36	0.36	0.35

* Mass of the same product (BASF MasterGlenium ACE-30) which is a solution containing 70% of water.

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
