# Peer review of "Mastering Yield Stress Evolution and Formwork Friction for Smart Dynamic Casting"

_materials, 2020, doi:10.3390/ma13092084_

Round 1

Reviewer 1 Report

The paper deals with the problems of a specific digital slip-forming process.
The strength evolution of different material compositions are studied in details.
Friction between the formwork walls and the concrete is investigated both theoretically and experimentally.
The link between the material properties, the process conditions and the designed geometry is also discussed.

In my opinion, in the paper the appropriate connection between material and mechanical engineering is shown and in addition the work has a direct implementation of the results in real technological problems.
It should be emphasized also that dr E. Lloret Fritschi and authors of the paper are experts in the digital slip-forming process.

Author Response

Thank you for your comments and your confidence in the quality of the paper.

The challenges experienced during the empirical investigation with Smart Dynamic Casting provided questions that guided our analysis. The importance of set on demand material processing, that has relevance for many other digital concrete fabrication methods, could also be highlighted.

We are pleased to read that the paper could clarify links between the theoretical framework and fabrication strategies with different formwork geometries.

Reviewer 2 Report

For 3D printing technology in construction domain, SDC with concrete is one of promising direction. This article conducted three parts of experiments in terms of materials using during concrete printing. The formula and experiment results are explained in details with clear diagrams.

For reader, the methods conducted might get a better understanding for the experiments with design examples in the discussion.

Author Response

Thank you for your valuable comments.

We inserted an additional figure (Figure1) with references at the beginning of the paper to illustrate the different type of architectural elements that were produced with SDC until now. This figure might also support the discussion of processing strategies.

We are aware that SDC is not the most common digital concrete process and it needs a detailed explanation for better understandability. However, we also think that this process deserves attention due to its potential to integrate conventional reinforcement, provide good surface quality and lower the risk of cold joint formation.

Reviewer 3 Report

The manuscript entitled “Mastering Yield Stress Evolution and Formwork Friction for Smart Dynamic Casting” reported the smart dynamic casting and introduce the approach to deal with different materials and mix design regarding friction and yield stress of materials.

Title

ok

Abstract

Line 21: “process window” what is the process window? It is better to describe it.

Keywords:

The number of keywords is too much.

Introduction

Line 35-36: Please revise the sentence, grammatically is not correct.

Line 36: there are few other studies on rheology

Line 37-38: this sentence “ that uses …than the elements” it does not make sense please and write in a way reader understand.

Line 39: when you are talking about the robotic arm axis and mixed up with the linear axis, please write precisely because this is confusing and many studies on the robotic arm work that have been done on that, please do not only cite on a specific community, here are few that have been done work on the path of robotic arm;

https://www.sciencedirect.com/science/article/pii/S0950061819325176?casa_token=jqIuOmEr4IkAAAAA:V3HTcAi3BDTdwOYkUcqB5LHN-zfS96TMNF0YgDgRRv0bMtnKg9CLp5fxWDOdsX_Mg7AcsEMq-3Q

line 39-40: if the formwork has been removed then why are we using it? Can authors reply this because it just increases the time of fabrication and I feel we are casting like a conventional method?

Line 41: what do you of “ known rate”? it is vague and out of telling the whole story? Do you mean casting rate?

Line 42: while authors used retarded and assist open time, please cite previous work. Unfair to mention some and ignore others.

https://link.springer.com/article/10.1617/s11527-012-9828-z

line 43: how the acceleration is happened by adding admixture or physical action, please revise. Authors statement is not clear.

Line 44-45: it has a grammatical mistake, please revise.

Line 46: it has the issue of the grammar again. Please revise.

Line 59: please support your evidence by references” less wasteful and less costly”.

Line 59: using the first pronoun in academic writing is not professional.

Line 75-78: this aim is not clear in terms of connection with the previous goal. Please revise in better words.

Please read these work as well, there are many others, then comprehend and collect them. Authors should explain, why have been chosen this title.

Materials and Methods

2.1 materials

2.1.1

Line 88-90: is the retarded base same as in the mix of Self-compacting concrete and Ultra High-Performance Concrete ? please revise and explain.

Line 88-90:”UHPFRC” do you mean of “ Ultra High-Performance Fibre Reinforced Concrete” please write full name before abbreviation.

Line 99: what do you mean of “ force action mixer”?

Line 118: what do you mean of section 0?? Please revise

Line 119: mix designs used in the mortars not“  used for the mortars”

Table 1. what type of Glass fibre has been used ? for more information refer to previous work of other researcher and why? Because there are too many types and glass fibre usually affect by alkali of cementitious materials causes damage on if we are not using the right type.

https://www.sciencedirect.com/science/article/pii/S0958946505000545?casa_token=1w9xZTPyA68AAAAA:XruGYVyT5Otmh_GYC5bAvDQnP5450E1RS916-y97PXsJJcGr1FhlvTn_1QCw2SENrWj9eGwFNkc

line 127: please write the exact type of superplasticizer.

Line 139: section 0?? Where is that section?

Table 2: the caption of the table does not caption. It is a paragraph should know the difference between caption and paragraph. Use subscript for a chemical compound. Revise

2.2 Methods

2.2.1

Line 152: please refer to previous work that has been used mini-cone for similar purposes;

https://journals.sagepub.com/doi/full/10.1177/0361198120907583

2.2.4

Line 175: why the strip dimension are like that ? is that been design it previously for such work because the thickness of 0.5 mm is too low for such work.

Line 185: previously authors have been mentioned mortar, now is written concrete. These two terms are different. Just wonder we are casting walls out of concrete, not mortar.

Line 197: is there any guideline or standard to follow to pull out the formwork at the certain speed. Please explain.

Figure 2: again the caption is too long to read. I prefer authors to explain some of the terminologies on the figure than put all in the caption.

Figure 6, 7 and 8: most of the peak have appeared at 10 h. Authors should explain what is the reasons.

3.2 disccusion

Line 320, 324: again mentioned concrete. please revise

Line 344, 345, 347: concrete or mortar. Revise?

Line 362: whether say w/c ratio or w/b ration be consistent in the entire paper.

Line 371?: section 0 ?

Line 374: “it will probably … drying shrinkage” how could be? maybe plastic shrinkage or chemical shrinkage because authors focus at the early age of mortar nor the late stage of drying. Revise

Figure 9: pull-out stress, might be not right term there will be confused later with pull-out test and pull-out stress of bolt and steel in the concrete. revise

Line 392: again used term concrete, I am not sure whether the authors used concrete or mortar. And if it is mortar why? Because the mortar is not used to build a wall.

Line 412 to 420: this paragraph should come before the figure.

Line 412 to 420: if there are such differences, please explain what is the reason. Is it just using admixtures or pumping rate, or waiting time?

4.2 Discussion

Line 426: using the pull-out test is the wrong term? It will mix with the pull-out test for the steel in the concrete. I recommend replacing this terminology with lift-off test or slip off test in the entire of the paper.

Line 444; that is not a professional way to write”by …in her PhD”. Please rewrite the sentence.

Line 467-518: there is not any consideration regarding the movement of the robotic arm to the attached formwork. As mentioned in the author’s work, they have been used very thin formwork which is possible to buckle or oscillation happens at the end of the robotic arm. Explain

Line 610-611: we usually say calcium silicate hydration not “tricalicium…”? revise. If say tricalcium silicate that is ok them or mean C-S-H.

Line 609-610: how authors prove that there is no sulfate depletion?

Line 618: using aluminium sulfate two times is very unique in writing. Please revise

Line 620-622: that sentence may be controversial. Especially if have sulfate it maybe a good option to use, why not used calcium aluminate cement. Another point, aluminate helps to set quickly but at a later age the strength lower than normal product.

Overall, the paper is well written but it needs a major revise prior to be published in the journal.

Round 2

Reviewer 3 Report

Now, paper presented in a better format.